# *Aspergillus flavus* and *Fusarium verticillioides* and Their Main Mycotoxins: Global Distribution and Scenarios of Interactions in Maize

**DOI:** 10.3390/toxins15090577

**Published:** 2023-09-18

**Authors:** Xiangrong Chen, Mohamed F. Abdallah, Sofie Landschoot, Kris Audenaert, Sarah De Saeger, Xiangfeng Chen, Andreja Rajkovic

**Affiliations:** 1Department of Food Technology, Safety and Health, Faculty of Bioscience Engineering, Ghent University, 9000 Ghent, Belgium; mohamed.fathi@ugent.be (M.F.A.); andreja.rajkovic@ugent.be (A.R.); 2Department of Plants and Crops, Faculty of Bioscience Engineering, Ghent University, 9000 Ghent, Belgium; sofie.landschoot@ugent.be (S.L.); kris.audenaert@ugent.be (K.A.); 3Department of Forensic Medicine and Toxicology, Faculty of Veterinary Medicine, Assiut University, Assiut 71515, Egypt; 4Centre of Excellence in Mycotoxicology and Public Health, Department of Bioanalysis, Faculty of Pharmaceutical Sciences, Ghent University, 9000 Ghent, Belgium; sarah.desaeger@ugent.be; 5Department of Biotechnology and Food Technology, Faculty of Science, University of Johannesburg, Doornfontein Campus, P.O. Box 17011, Gauteng 2028, South Africa; 6Shandong Analysis and Test Centre, Qilu University of Technology (Shandong Academy of Science), Jinan 250014, China; xiangfchensdas@163.com

**Keywords:** food safety, maize, *Aspergillus flavus*, *Fusarium verticillioides*, co-occurrence, aflatoxin B1, fumonisin B1

## Abstract

Maize is frequently contaminated with multiple mycotoxins, especially those produced by *Aspergillus flavus* and *Fusarium verticillioides*. As mycotoxin contamination is a critical factor that destabilizes global food safety, the current review provides an updated overview of the (co-)occurrence of *A. flavus* and *F. verticillioides* and (co-)contamination of aflatoxin B1 (AFB1) and fumonisin B1 (FB1) in maize. Furthermore, it summarizes their interactions in maize. The gathered data predict the (co-)occurrence and virulence of *A. flavus* and *F. verticillioides* would increase worldwide, especially in European cold climate countries. Studies on the interaction of both fungi regarding their growth mainly showed antagonistic interactions in vitro or in planta conditions. However, the (co-)contamination of AFB1 and FB1 has risen worldwide in the last decade. Primarily, this co-contamination increased by 32% in Europe (2010–2020 vs. 1992–2009). This implies that fungi and mycotoxins would severely threaten European-grown maize.

## 1. Introduction

Maize (*Zea mays* L.) is one of the strategic cereal crops which can be processed into a variety of food, feedstuff, and other industrial products [1]. The threat posed to maize production by fungal plant diseases is one of the critical factors that can destabilize global food security and safety. Preharvest losses due to fungal plant diseases are estimated to account for nearly 10–20% of cultivated maize, which can feed about 8.5% of the world’s population [2]. Among these diseases, *Aspergillus* Ear Rot and *Fusarium* Ear Rot, caused by *Aspergillus* and *Fusarium* species, respectively, are the most important [3]. Both diseases decrease the yield and quality of the maize crop and the safety of maize kernels due to the production of mycotoxins, secondary fungal metabolites toxic to animals and humans [4].

*Aspergillus* Ear Rot disease is mainly caused by a fungal pervasive maize invader called *A. flavus* [5]. The *A. flavus* species has been reported in several countries in Africa, America, Asia, and Europe [6,7,8]. Toxigenic *A. flavus* species produce several mycotoxins/secondary metabolites; however, due to their toxicity and widespread contamination, the most studied toxins are aflatoxins (AFs) [9]. So far, there are four members of AFs called B1, B, G1, and G2. Aflatoxin B1 (AFB1) is the most potent member of AFs, and several fatal outbreaks have been associated with the consumption of AFB1-contaminated maize in Brazil (60 deaths) and Kenya (317 cases of intoxications and 125 deaths) [10,11]. The toxicity of AFB1 has aroused widespread public concern due to its hepatotoxic, immunotoxic, mutagenic, carcinogenic, and teratogenic properties [12]. The International Agency for Research on Cancer (IARC) classified AFB1 as a group 1 carcinogen due to the sufficient evidence of causing liver cancer in humans [13].

The other common fungal disease in maize is *Fusarium* Ear Rot, which is mainly caused by *Fusarium verticillioides* [14,15]. Similarly to *A. flavus*, the *F. verticillioides* species has been reported as a worldwide fungal pathogen of maize kernels. The fungus is also considered a major producer of an important group of mycotoxins called fumonisins (FBs) [16]. There are three members of FBs called fumonisin B1, FB2, and FB3; however, the main member of FBs is FB1 (FB1) [17]. Several foodborne outbreaks due to consumption of FB1-contaminated maize were reported over the years in South Africa (45 cases), Mexico (>100 cases), India (1424 cases), and Brazil (66 cases) [18,19,20]. Several studies have shown that FB1 can pose many toxic effects (neurotoxic, hepatotoxic, and nephrotoxic) in humans. The IARC classified FB1 as a class 2B carcinogen (possible human carcinogen) [21,22].

Given the significant negative impacts of these fungi and their mycotoxins on agriculture and human health, this review focuses on the (co-)occurrence of *A. flavus* and *F. verticillioides* and the (co-)contamination of AFB1 and FB1 in maize. Furthermore, it highlights how *A. flavus* and *F. verticillioides* interact with each other in maize.

## 2. Global Distribution of *A. flavus* and *F. verticillioides* in Maize

A general overview of the number of studies per country reporting the (co-)occurrence of *A. flavus* and *F. verticillioides* in different continents between 1980 and 2020 is shown in Figure 1. Furthermore, their sampling years, sample numbers, and the percentage of occurrence are shown in Table 1. Most of these studies were on *A. flavus* occurrence, followed by *F. verticillioides*, while fewer studies on the co-infection were published. The research of *A. flavus* was highest in Africa, with 74 scientific papers, followed by Asia (39 studies), Europe (35 studies), and the Americas (14 studies). However, Europe has a high awareness of studying the contamination of *F. verticillioides* that was reported in 51 papers, followed by Africa (40 studies), Asia (37 studies), and the Americas (27 studies). These data show that people are paying more attention to the contamination of both fungi [4], which reflect that the mycotoxin pollution problem of these two is increasing during these years. Figure 1 shows that more research of *A. flavus* and *F. verticillioides* is related to hot and rainy climates in African countries, which favored the growth of the two fungi. Apart from Africa, it was noticeable that southern European countries (Italy, Portugal, Spain, and Romania), some Asian countries (China, India, Iran, and Pakistan), and other countries in Latin and northern Americas (Brazil, Argentina, and United States) had a considerable number of publications. Gradually, those areas face an increased risk of *A. flavus* and *F. verticillioides* co-contamination [23]. In total, there are 30 papers in the literature reporting the co-occurrence of *A. flavus* and *F. verticillioides* in maize from all continents: Africa (12 studies); Europe (8 studies); Asia (6 studies); and the Americas (4 studies). It is noticeable that all these previous surveys were conducted in countries with hot climates. However, there is an increasing number of studies in European cold climate countries due to global warming. Global warming is no longer a trend but a reality because many countries have excessive emissions of CO_2_ [24]. To mitigate the threat of climate change, 195 countries agreed to limit the emission of CO_2_ by adopting new rules [25]. The co-occurrence of *A. flavus* and *F. verticillioides* will likely increase worldwide, especially in Europe.

**Table 1 toxins-15-00577-t001:** Reported (co-)occurrences of *A. flavus* and *F. verticillioides* in maize worldwide.

Country	*A. flavus*			*F. verticillioides*			Co-Occurrence of Both Fungi				Reference
	Sample		Occurrence of *A. flavus* (%)	Sample		Occurrence of *F. verticillioides* (%)	Sample		Occurrence of *A. flavus* (%)	Occurrence of *F. verticillioides* (%)	
	Year	Number		Year	Number		Year	Number			
**Africa**											
Benin	1994	80	80	2000	800	68	2000	800	48	68	[26,27]
	1995	60	60	2005	100	10	2005	100	30	10	[26,28]
	1994	400	74								[29]
	1995	300	56								[29]
	1996	88	90								[30]
	1994	625	20								[31]
	1995	625	47								[27]
	2000	800	48								[27]
	2005	100	30								[32]
	2009	60	+								[33]
	2018	50	76								[34]
Burkina Faso	2019	10	40								[35]
Cameroon	1997	72	1	1997	72	22	1997	72	1	22	[36]
	2005	95	53								[37]
Egypt	2003	90	80	1996	72	39	2012	40	33	3	[28,38,39]
	2012	40	33	2003	90	80	2012	50	41	27	[39,40]
	2012	50	41	2012	40	3					[34,39]
	2013	13	85	2012	50	27					[33,41]
Ethiopia	2015	30	37	1995	36	52					[42,43]
	2015	100	8	2012	200	42					[44,45]
	2016	150	64	2014	200	73					[46,47]
				2015	100	56					[44]
Ghana	2003	25	84								[48]
	2013	326	+								[49]
	2016	800	99								[50]
	2017	60	34								[51]
	2018	90	+								[52]
	2020	180	+								[7]
Kenya	2006	165	93	1996	197	60					[53,54]
	2006	156	58	2008	86	40					[55,56]
	2007	716	+	2010	985	+					[57,58]
	2010	513	78								[59]
	2012	113	79								[60]
	2012	629	39								[61]
	2013	300	86								[62]
	2015	514	25								[63]
	2017	120	78								[64]
	2018	120	67								[65]
Lesotho	2010	40	20	2010	40	17	2010	40	20	17	[66]
Liberia	2005	23	16	2005	23	15	2005	23	16	15	[67]
Malawi	2008	178	+								[68]
	2012	156	+								[69]
Niger	2012	39	10								[70]
Nigeria	1992	43	+	2001	103	89	2001	103	65	89	[71,72]
	2001	103	65	2003	27	67	2005	180	15	18	[72,73,74]
	2003	66	67	2004	103	51	2005	23	100	87	[48,67,75]
	2005	180	15	2005	180	18	2019	93	20	19	[73,76]
	2005	13	83	2005	182	70					[77,78]
	2005	260	+	2005	23	87					[67,79]
	2005	23	100	2006	50	82					[67,80]
	2007	55	85	2019	93	19					[76,81]
	2011	78	+								[82]
	2015	18	26								[83]
	2018	36	6								[84]
	2019	142	+								[85]
	2019	93	20								[75]
	2020	46	50								[86]
South Africa	2010	40	43	1997	211	16	2010	40	43	88	[87,88]
	2011	54	0.3	2000	211	16	2011	54	0.3	28	[88,89]
	2013	100	12	2003	211	32	2013	100	12	76	[88,90]
	2017	32	10	2003	44	+					[91,92]
				2006	140	19					[93]
				2007	114	10					[93]
				2009	45	52					[94]
				2010	54	70					[95]
				2010	40	88					[87]
				2011	54	28					[89]
				2013	100	76					[90]
				2018	24	92					[96]
Tanzania	2012	200	+								[97]
Togo	2015	55	+	2015	55	+	2015	55	+	+	[98]
	2018	70	+								[99]
Tunisia	2011	10	100								[100]
	2011	21	67								[101]
Zambia	2006	100	18								[102]
	2015	250	60								[103]
	2017	800	67								[104]
**Americas**											
Argentina	1998	50	78	1994	50	46	1998	50	78	42	[105,106]
	2000	100	70	1996	210	+					[107,108]
	2008	90	100	1996	51	45					[109,110]
	2014	40	73	1997	462	29					[111,112]
				1998	158	61					[113]
				1998	540	22					[114]
				1998	50	42					[105]
				2015	30	98					[115]
				2016	30	83					[115]
				2017	30	67					[115]
Brazil	1995	66	15	1991	48	85	1995	66	15	61	[116,117]
	1998	110	+	1995	66	61	2005	200	12	86	[116,118,119]
	1999	60	64	1998	56	+	2008	464	80	40	[120,121,122]
	2003	121	+	1998	87	+					[122,123]
	2004	20	100	2005	200	86					[119,124]
	2005	200	12	2008	464	40					[119,121]
	2008	464	80	2010	200	+					[121,125]
	2012	200	38								[126]
Canada				1980	100	+					[127]
Costa Rica	1992	100	70								[128]
Honduras	1993	52	6	1993	52	71	1993	52	6	71	[129]
Mexico	1995	87	75	2001	28	+					[130,131]
	2006	83	+	2003	160	65					[132,133]
United States	1996	15	+	1986	41	98					[134,135]
	2012	30	+	1998	100	50					[136,137]
	2017	283	12	1999	40	+					[138,139]
				2000	120	57					[140]
				2001	50	+					[141]
				2005	818	+					[142]
Venezuela	1993	37	+	1993	37	+	1993	37	+	69	[143]
				1998	79	69					[144]
**Asia**											
China	1998	40	+	1998	40	+	1998	40	+	+	[145]
	2003	120	99	2005	64	+	2008	87	+	+	[146,147,148]
	2008	87	+	2008	87	+	2014	44	52	25	[147,149]
	2014	44	52	2011	307	+					[149,150]
	2014	105	95	2012	362	62					[151,152]
				2012	146	+					[153]
				2012	250	18					[154]
				2012	225	11					[155]
				2013	225	19					[155]
				2013	175	30					[156]
				2014	225	19					[155]
				2014	44	25					[149]
				2019	110	+					[157]
India	1987	400	19	2007	43	22	2012	150	85	60	[158,159,160]
	1995	2074	+	2011	15	67	2013	45	16	84	[161,162,163]
	1997	197	60	2012	150	60					[159,164]
	2009	38	82	2013	45	84					[162,165]
	2011	660	40	2014	53	3					[166,167]
	2011	32	+	2015	106	90					[168,169]
	2011	106	57								[170]
	2012	150	85								[159]
	2013	45	16								[162]
	2013	86	56								[171]
	2016	595	+								[172]
Indonesia	1995	16	75	1995	16	50	1995	16	75	50	[173]
Iran	2000	92	6	2000	92	52	2000	92	6	51	[174]
	2000	51	+	2004	41	60					[175,176]
	2011	54	+	2009	460	+					[177,178]
	2011	160	44	2016	182	59					[179,180]
Korea				2009	19	70					[181]
Malaysia	2009	80	87	2008	398	14					[182,183]
				2009	80	47					[182]
Nepal				1997	78	85					[184]
Pakistan	2007	90	26	2007	90	10	2007	90	26	10	[185]
	2007	100	70								[186]
	2007	36	+								[187]
	2007	65	+								[188]
	2008	40	+								[189]
	2010	18	94								[190]
	2013	100	+								[191]
	2016	45	+								[192]
	2017	57	+								[192]
	2018	155	+								[192]
	2019	67	+								[192]
Saudi Arabia	2013	40	50	2013	40	32					[193]
				2014	60	63					[194]
Vietnam	2000	45	31	1996	50	+	2005	25	92	23	[195,196]
	2005	25	92	2005	25	23					[195]
	2009	102	29	2019	93	47					[197,198]
Yemen	2016	20	30	2016	20	12	2016	20	30	12	[199]
**Europe**											
Belgium				2017	900	0.4					[200]
				2017	257	99					[201]
				2017	257	54					[201]
Croatia	1993	90	8								[202]
	2014	50	+								[203]
France	2015	225	68	1999	72	73					[204,205]
Germany	2006	44	82								[206]
	2017	180	13								[207]
	2018	113	39								[207]
Hungary	2010	104	64								[208]
	2014	20	+								[203]
	2014	196	26								[209]
Italy	2002	280	+	1993	600	100					[210,211]
	2003	70	62	2007	83	53					[6,212]
	2003	280	+	2008	90	100					[210,213]
	2004	354	+	2010	30	95					[210,214]
	2005	354	+	2010	50	42					[210,215]
	2006	354	+	2011	39	37					[210,214]
	2007	83	+	2011	140	+					[211,216]
	2010	134	46	2017	46	22					[217,218]
	2010	30	4	2018	46	13					[214,218]
	2011	140	+	2020	177	47					[216,219]
	2011	39	1								[214]
	2017	46	23								[218]
	2018	46	12								[218]
	2020	177	57								[219]
Poland				2011	30	93					[220]
				2014	100	47					[221]
				2015	83	35					[221]
				2016	58	35					[221]
				2017	48	39					[221]
Portugal	2011	22	9	2005	30	+					[222,223]
				2005	67	22					[224]
				2005	31	+					[225]
				2018	9	80					[226]
Romania	2004	54	33	2004	54	18	2004	54	33	18	[227]
	2008	42	43	2008	42	7	2008	42	43	7	[228]
	2009	32	32	2009	32	+	2009	32	32	+	[228]
	2010	12	67	2010	12	17	2010	12	67	17	[228]
Serbia	2012	180	+	2010	-	+	2012	200	12	34	[229,230,231]
	2012	200	12	2012	200	34	2012	29	37	15	[230,232]
	2012	29	37	2012	29	15					[232]
	2014	80	+	2012	90	+					[203,233]
	2015	180	+	2018	18	9					[234,235]
	2017	458	+								[236]
Slovakia				1996	550	50					[237]
				1998	550	43					[237]
Spain	2004	54	33	1996	55	91	2009	60	43		[227,238,239]
	2009	60	43	1999	48	60	2014	49	27		[205,238,240]
	2014	49	27	2003	60	12	2018	27	82		[240,241,242]
	2018	27	82	2004	54	18					[205,241]
				2009	60	50					[238]
				2014	49	100					[240]
				2018	27	52					[241]
Switzerland				2006	420	46					[243]
				2010	17	16					[244]
				2010	289	+					[245]
United Kingdom				2012	990	1					[246]

+: occurrence without percentage. The occurrence percentage of *A. flavus* or *F. verticillioides* alone or together in maize per continent between 1980 and 2020 is depicted in Figure 2 and their sampling years, sample numbers, and the percentage of occurrence are shown in Table 1. Over the period between 1980 and 2020, there was a considerable variation in the occurrence percentage for *A. flavus* and *F. verticillioides* in maize in all the continents, which does not provide a consistent trend. Such variation shows that predicting the contamination of these fungi is difficult. Indeed, co-founding factors such as sample size, sampling strategies, fungal isolation, and identification methods affect the reported results in these papers. Comparing the median values for *A. flavus* and *F. verticillioides* occurrence percentages among the four continents shows that the occurrences in Europe are the lowest. On the other hand, the Americas (North and Latin America) had the highest occurrence percentages for both fungi in the surveyed maize samples (Figure 2).

This matches with the increasing awareness of global warming, which is expected to impact the presence of mycotoxins in food and feed severely. Battilani et al. reported that AFB1 is predicted to become a food safety issue in maize cultivated in Europe, especially under the +2 °C scenario, the most probable climate change scenario for the following years [23,247]. A similar scenario applies to *F. verticillioides* and FB1 in maize [104]. However, after considering the publications focused on isolating both fungi in maize, it is seen that the median values for the occurrences of both fungi in Africa and Europe are close. Different overview of the occurrence percentages for both fungi in America in which the median value for the occurrence percentage of *A. flavus* is three times higher than *F. verticillioides* which is the opposite situation in Asia. This also shows the significant variation in the detection of both fungi in maize samples and the difficulty in drawing a consistent conclusion.

On the other hand, researchers investigated the possible interactions between *A. flavus* and *F. verticillioides* and their toxins in maize, which is presented in the following section of this review. Furthermore, a global summary is provided on the (co-)occurrence of the commonly produced mycotoxins (AFB1 and FB1) in maize.

## 3. Worldwide Co-Occurrence of AFB1 and FB1 in Maize and Maize-Based Products

The simultaneous occurrence of several mycotoxins in a single product is a common situation, with the natural co-contamination of AFB1 and FB1 in maize and maize products as an example. An overview of the surveys conducted on AFB1 and FB1 is summarized in Table 2, which contains the sampling years, sample numbers, detection methods, and concentrations of both toxins between 1991 and 2020. The most common analytical technique (up to 66.7%) used for detecting and quantifying AFB1 and FB1 in the last decade was liquid chromatography–tandem mass spectrometry (HPLC-MS/MS). This is owing to the essential strengths of HPLC-MS/MS, including potentially high analytical specificity, a wide range of applicability to small and large molecules, the capability of multi- and mega-parametric tests, and the opportunity to develop robust assays with a high degree of flexibility within a short time frame [248].

In Africa, high concentrations of FB1 were 10,447 μg/kg and 18,184 μg/kg, and AFB1 concentrations that co-occur with high FB1 were 6738 μg/kg and 1081 μg/kg, respectively (see Table 2) [249,250]. Based on this, it was found that there can be a positive relationship between AFB1 and FB1 under this co-existence condition with the collected data in Africa: the concentration of AFB1 is correspondingly high/low in the presence of high/low concentrations of FB1 according to the correlation coefficient (r > 0.8). However, Sangare-Tigori et al., Kpodo et al., and Kimanya et al. contradicted this positive relationship, which can be the selection of detection methods. In the Americas, 70% of FB1 were higher than 2000 μg/kg, and the highest was up to 53,000.0 μg/kg [251], almost ten times more than in Africa under the co-occurrence of AFB1 and FB1. In Asia, the highest concentrations of FB1 and AFB1 were 37,000 μg/kg and 4030 μg/kg in the analyzed samples [18,252]. Moreover, since 2010, AFB1 concentration was significantly decreased compared with before 2010. However, there was no apparent interaction between AFB1 and FB1 in the samples in the Americas (r < 0.1) and Asia (r < 0.1). In Europe, with the co-occurrence of AFB1 and FB1, FB1 contamination was severe, and 85.7% of cases exceeded 2000 μg/kg. There were even 57.1% of cases higher than 10,000 μg/kg; the highest was up to 51,690 μg/kg [253]. However, AFB1 concentrations are lower than in other continents, and the increase in FB1 hardly affects AFB1 concentrations under their co-existence. It was found that the co-occurrence of both toxins was detected in Serbia and Spain by 2012, which can be a portent of the co-contamination of AFB1 and FB1 threatening to Europe [254,255].

**Table 2 toxins-15-00577-t002:** Reported co-occurrence of aflatoxin B1 (AFB1) and fumonisin B1 (FB1) in maize and maize production worldwide.

Country	Sample		Method of Detection	FB1 (μg/kg)			AFB1 (μg/kg)			Reference
	Year	Number		Min	Mean	Max	Min	Mean	Max	
**Africa**										
Côte d’Ivoire	2006	10	ELISA	300.0	900.0	1500.0	1.5	4.1	20.0	[256]
Egypt	2012	40	HPLC-FLD	12.0	171.0	947.0	0.2	3.7	19.2	[39]
	2015	79	HPLC-MS/MS	1.0	68.0	2453.0	0.3	4.8	197.5	[257]
Ghana	2000	15	HPLC-FLD	11.0	358.0	655.0	0.0	54.5	204.0	[258]
Malawi	2016	90	HPLC-FLD	100.0	900.0	7000.0	0.7	8.3	140.0	[259]
Nigeria	2019	69	HPLC-MS/MS	390.0	589.0	765.0	1.4	9.1	27.9	[260]
	2011	103	HPLC-FLD	70.0	495.0	1870.0	3.0	22.0	130.0	[72]
	2012	70	HPLC-MS/MS	1.8	1552.0	10,447.0	0.4	394.0	6738.0	[261]
Tanzania	2008	120	HPLC-FLD	144.0	206.0	363.0	5.0	51.0	90.0	[249]
	2015	60	HPLC/TOFMS	16.0	1361.0	18,184.0	2.0	65.0	1081.0	[262]
	2017	7	HPLC-FLD	57.0	329.0	1672.0	0.5	1.3.0	364.0	[250]
Zimbabwe	2016	388	HPLC-FLD	10.0	476.0	607.0	0.6	3.2	26.6	[263]
	2016	95	HPLC-MS/MS	<12.5	242.0	1106.0	<3.8	11	11.0	[264]
**Americas**										
Argentina	1995	4000	HPLC-FLD	173.0	578.0	1935.0	4.0	5.0	6.0	[265]
Brazil	2010	214	HPLC-FLD	200.0	2200.0	6100.0	0.2	9.4	129.0	[266]
	2004	200	HPLC-FLD	15.0	1773.0	9670.0	6.8	29.1	1393.0	[119]
	2016	26	HPLC-MS/MS	17.0	350.0	53,000.0	8.7	100.0	390.0	[267]
	2020	186	HPLC-MS/MS	n.r.	3270.0	n.r.	n.r.	1.5	n.r.	[268]
	2008	24	HPLC-FLD	157.0	2940.0	9707.0	0.5	2.6	38.0	[251]
	2001	150	IC-ELISA	96.0	5080.0	22,600.0	38.0	191.0	460.0	[269]
Guatemala	2012	640	HPLC-MS	0.0	1800.0	17,100.0	0.0	63.0	2655.0	[270]
United States	2003	7	ELISA	4.0	74.0	263.0	0.1	0.8	1.5	[271]
Venezuela	1993	37	HPLC-FLD	25.0	1486.0	15,050.0	0.0	4.5	50.0	[143]
**Asia**										
China	2011	108	HPLC-UV	0.0	1247.0	37,000.0	0.4	6.5	136.8	[272]
	2011	51	HPLC-MS/MS	1.0	325.0	1997.0	0.1	1.1	2.1	[273]
	2016	203	HPLC-MS/MS	10.0	30.5	255.0	1.5	1.8	2.3	[252]
	1998	40	HPLC-FLD	58.0	377.0	1796.0	9.0	460.0	2496.0	[145]
India	1997	35	HPLC-reverse	10.0	620.0	4740.0	0.1	2.6	4030.0	[18]
	2013	45	TLC-UV	49.6	155.3	650.0	20.6	161.3	402.4	[162]
Indonesia	1995	16	GC-MS	51.0	788.0	2440.0	4.0	102.0	428.0	[173]
	1994	12	HPLC-FLD	226.0	843.0	1780.0	1.0	352.0	3300.0	[274]
Iran	2009	35	HPLC-UV	n.r.	5820.0	n.r.	n.r.	9.5	n.r.	[275]
Korea	2017	507	HPLC-MS/MS	4.0	137.0	2990.0	0.0	5.2	5.2	[276]
	2002	47	HPLC-FLD	43.0	74.0	119.0	14.0	20.0	25.0	[277]
Philippines	1994	50	HPLC-FLD	57.0	491.0	1820.0	1.0	49.0	430.0	[274]
Thailand	1992	18	HPLC-FLD	63.0	1790.0	18,800.0	1.0	72	606.0	[278]
	1994	27	HPLC-FLD	63.0	1580.0	18,800.0	1.0	63	606.0	[274]
Türkiye	2003	19	ELISA	1.0	88.0	367.0	0.0	10.9	32.3	[271]
Vietnam	1993	32	HPLC	n.r.	1101.0	n.r.	n.r.	28	n.r.	[279]
	2005	25	HPLC-FLD	400.0	1121.0	3300.0	2.1	21.8	126.5	[195]
**Europe**										
Croatia	2007	24	ELISA	200.0	7630.0	20,700.0	2.7	3.4	4.5	[280]
Italy	1995	98	HPLC-UV	55.0	3347.0	51,690.0	0.1	1.9	109.0	[281]
	1996	104	HPLC-UV	53.0	1324.0	7285.0	0.1	0.3	13.0	[281]
	1997	94	HPLC-UV	72.0	3103.0	47,078.0	0.1	1.5	32.0	[281]
	1998	114	HPLC-UV	55.0	2655.0	13,763.0	0.1	1.5	28.0	[281]
	1999	93	HPLC-UV	54.0	5173.0	21,132.0	0.1	4.1	128.0	[281]
Serbia	2013	127	ELISA	0.0	2363.0	10,860.0	0.0	18.5	491.7	[282]
	2012	9	ELISA	80.0	358.0	1220.0	0.0	6.2	26.3	[283]
	2012	200	ELISA	880.0	1611.0	2950.0	0.3	1.4	2.4	[282]
	2012	51	HPLC-MS/MS	211.0	4121.0	13,396.0	0.6	44.0	205.0	[282]
	2013	51	HPLC-MS/MS	88.0	4690.0	16,187.0	0.5	8.0	48.0	[282]
	2014	51	HPLC-MS/MS	193.0	5846.0	27,103.0	0.0	0.1	0.3	[282]
	2015	51	HPLC-MS/MS	192.0	1905.0	4253.0	0.4	8.0	41.0	[282]
Spain	2016	148	HPLC-MS/MS	99.0	287.0	857.0	<0.1	1.2	8.5	[253]
	2015	10	HPLC-MS/MS	43.0	920.0	3754.0	<0.3	0.9	0.9	[255]
	2016	22	HPLC-MS/MS	28.0	8332.0	34,600.0	1.4	1.6	1.9	[255]
	2017	26	HPLC-MS/MS	26.0	7715.0	50,900.0	22.0	73	124.1	[255]
	2018	21	HPLC-MS/MS	40.0	2657.0	17,100.0	0.9	40.6	80.7	[255]
	2019	19	HPLC-MS/MS	29.0	920.0	3841.0	0.0	0.9	0.9	[255]
United Kingdom	1992	50	HPLC-FLD	6.0	1337.0	4550	1.0	4.9	41	[284]
**Australia**										
Australia	2010	1648	HPLC-UV	506.0	19,278.0	19,278.0	13.9	46.0	4278.0	[285]

Min: Minimum; Max: Maximum; HPLC: High-performance liquid chromatography; UV: Ultra-violet; FLD: Fluorescence detector; MS/MS: Mass spectrometry; IC: Indirect competitive; ELISA: Enzyme-linked immunosorbent assay; TOFMS: Time-of-flight mass spectrometry; TLC: Thin layer chromatography; n.r.: no report.

The mean of AFB1 and FB1 levels in studies from different continents is shown in Figure 3. As more than 70% of the produced maize was primarily used for animal feed in the world [286], the EU maximum limits for feed maize FB1(2000 µg/kg) and AFB1 (20 µg/kg) were selected as thresholds to interpret the collected data. From 1991 to 2020, 38% of AFB1 and 61% of FB1 studies exceeded the EU maximum limits separately. However, these excess issues have not happened in all continents under the co-occurrence of AFB1 and FB1. In Africa, the co-occurrence of both mycotoxins has risen to 53.8% since 2012. From 2012, 30.0% of survey studies are out of the AFB1 threshold, but all cases are below the FB1 threshold. In the Americas, 44.4% of AFB1 was higher than 20 µg/kg, and 33.3% of FB1 was higher than 2000 µg/kg. In Asia, 62.5% of studies exceeded the AFB1 limit, and only one study reported FB1 contamination exceeding the FB1 limit. There were no cases exceeding the EU maximum limits for both toxins in the last decade year. In Europe, the co-contamination of AFB1 and FB1 has increased to 31.4% since 2012. Over the period 2012 until 2020, 25.0% of AFB1 was beyond its limit, which never happened before 2012, and 58.3% of FB1 was beyond its limit, which decreased to 13.1% compared with before 2012. UK Climate has reported that the most recent decade (2011–2020) has been, on average, 0.5 °C warmer than the 1981–2010 average, and the 21st century so far has gradually been warmer, which is roughly consistent with the observed rate of global mean temperature warming [287]. Therefore, it can predict that the co-contamination of AFB1 and FB1 will become more serious worldwide due to global warming, and the risk of human co-exposure to both toxins will increase.

## 4. Interactions between *A. flavus* and *F. verticillioides* and Their Toxins in Maize

The outcome of the interactions between *A. flavus* and *F. verticillioides* differs depending on the applied laboratory conditions for each experiment. This includes the substrate, culture media (in vitro) or maize (in vivo), and the related incubation conditions. Fakhrunnisa and Ghaffar have proved that *A. flavus* inhibited the growth of *F. verticillioides* (inhibition rate 16.67%) by producing a zone of inhibition in the dual agar culture plate assay [288]. In case the incubation conditions are changed (e.g., temperature, CO_2_, and humility), the interaction between *A. flavus* and *F. verticillioides* can also change, as reported by Camardo Leggieri et al. [289]. In their study, the growth of *A. flavus* was affected by the co-inoculum of *F. verticillioides*, and colony diameter was significantly lower than that measured in pure colonies if the incubation was between 20 °C and 25 °C. On the contrary, at 35 °C, *A. flavus* growth was enhanced by the presence of *F. verticillioides* [289]. Consistently, Giorni et al. reported that the co-existence of *A. flavus* and *F. verticillioides* was influenced by the temperature and water activity [290]. They reported that with the presence of both fungi, *F. verticillioides* nutritionally dominated all the strains of *A. flavus* at 20 °C and 0.95 aw, while *A. flavus* always nutritionally dominated *F. verticillioides* at 30 °C with either high aw (0.98 aw) or reduced aw (0.87 aw) [290]. In a recent study, the effect on fungal growth and the production of their main mycotoxins (AFs and FBs) on co-inoculation were also investigated by another group [291]. It was demonstrated that the growth rate of *A. flavus* and *F. verticillioides*, when grown in dual or mixed culture, was slower compared with the growth rate in a single culture, and average growth rate reductions of 10% and 11% were observed for *A. flavus* and *F. verticillioides*, respectively. When *A. flavus* and *F. verticillioides* were mixed, the production of AFB1 and FB1 significantly decreased. Likewise, Lanubile et al. showed that in the co-occurrence of *A. flavus* and *F. verticillioides*, both mycotoxins resulted in the reduction compared with the amount produced with single inoculation, and these findings were independent of temperature [292].

The interaction between *A. flavus* and *F. verticillioides* under in vivo environment is also highly dynamic. It depends on the experimental conditions, the variable measured, and how they colonize the host. Chen et al. observed the symptoms of the lesion and mycotoxin production to evaluate the interaction of *A. flavus* and *F. verticillioides* in maize [290]. The dual inoculation resulted in reduced lesions of *A. flavus*. In contrast, the lesion size and toxin production of *F. verticillioides* were unaffected in the presence of *A. flavus* in maize at 25 °C. In contrast, their mixed inoculation resulted in more extensive lesions than a single *A. flavus* inoculation and higher FB production than a single *F. verticillioides* inoculation [290]. The study indicates that *A. flavus* can be more affected by *F. verticillioides* in maize. A previous study underlined the different abilities of *A. flavus* and *F. verticillioides* to grow simultaneously on maize since they usually occupy different niches regarding carbon sources [290]. It is stated that *F. verticillioides* seems to be dominant because it can use more carbon sources at the lowest temperatures (15 °C) and the highest aw levels (> 0.95 aw), while *A. flavus* becomes dominant at higher temperatures (>25–30 °C) and dry conditions (0.87 aw) [290,293]. However, Lanubile et al. reported that in the co-occurrence of *A. flavus* and *F. verticillioides*, mycotoxin production has no significant differences among three different temperature regimes (20, 25, and 30 °C) for maize kernel contamination. However, FBs and AFs decreased compared with single inoculation at all the tested temperatures [292]. It is worth mentioning that Lanubile et al. tested maize kernels as the in vivo host, different from the above baby maize tested by Chen et al. [291], which can be the cause of different interactions of *A. flavus* and *F. verticillioides*. Overall, the interaction of *A. flavus* and *F. verticillioides* is manifested in the resistance to the growth of each other both in vitro and in vivo. At the same time, mycotoxin production is highly dependent on the temperature and the tested inoculation host.

## 5. Conclusions and Outlook

Throughout the last 30 years, the virulence of *A. flavus* and *F. verticillioides* and the co-occurrence of AFB1 and FB1 is also gradually contaminating Africa, the Americas, and Europe. There was no consistent trend for the co-occurrence of *A. flavus* and *F. verticillioides* in maize on all the continents. However, this co-occurrence is increasing in the world. In the last decade (2010–now), the co-contamination of AFB1 and FB1 has risen by 32% in Europe. It will threaten food safety and amplify food insecurity crises and increase the risk of co-exposure to both toxins for the public. Therefore, the (co-)occurrence of *A. flavus* and *F. verticillioides* pose significant concerns for co-contamination in the food, especially for the (co-)occurrence of the commonly produced mycotoxins (AFB1 and FB1) in maize. This (co-)occurrence would interact with the growth of both species and mycotoxin production, but more field data supporting their interaction are needed.

## Figures and Tables

**Figure 1 toxins-15-00577-f001:**
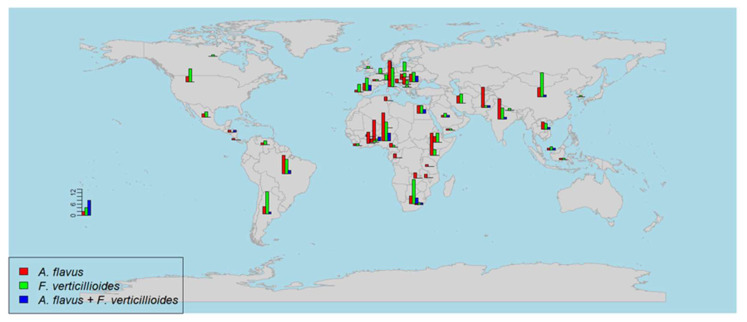
A world map showing the number of studies that surveyed the (co-)occurrence of *A. flavus* and *F. verticillioides*. The number of studies is represented as a bar chart for *A. flavus* (red color), *F. verticillioides* (green color), and both fungi (blue color).

**Figure 2 toxins-15-00577-f002:**
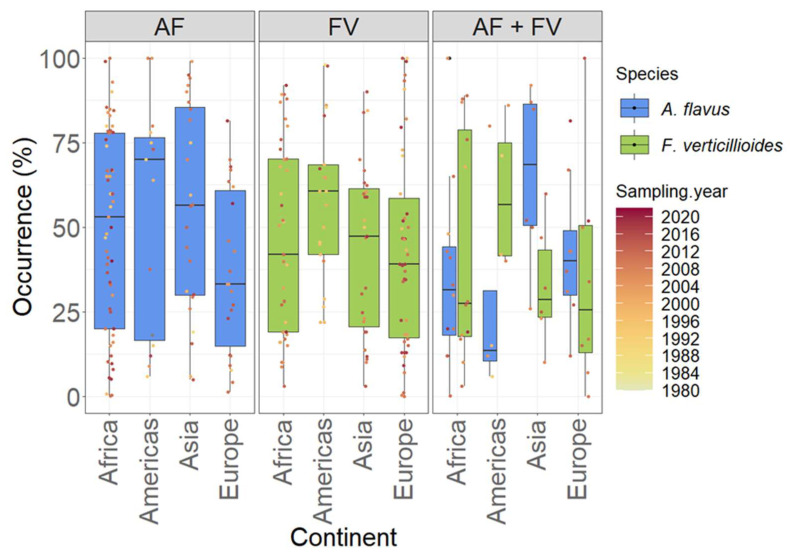
Boxplots show the percentage of contaminated maize samples with *A. flavus*, *F. verticillioides*, and both fungi in survey studies from Africa, the Americas, Asia, and Europe. The data points are colored according to the year of sampling.

**Figure 3 toxins-15-00577-f003:**
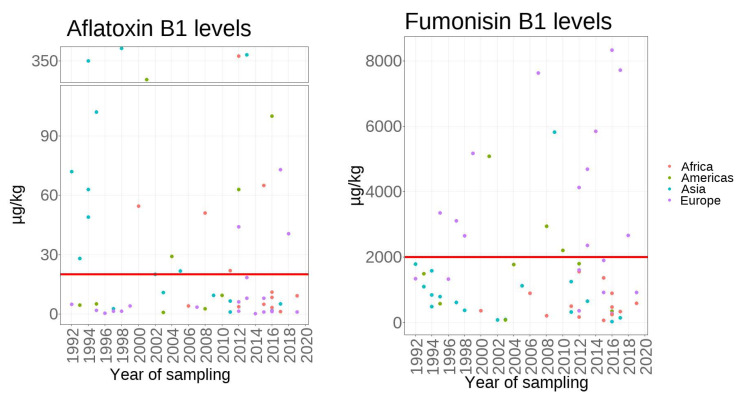
The reported mean values of aflatoxin B1 (AFB1) and fumonisin B1 (FB1) in studies from different continents. The red lines show the EU maximum limit for AFB1 (20 µg/kg) for cereals and FB1 (2000 µg/kg) for unprocessed maize.

## Data Availability

Not applicable.

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
