# Peer review of "Aspergillus flavus and Fusarium verticillioides and Their Main Mycotoxins: Global Distribution and Scenarios of Interactions in Maize"

_toxins, 2023, doi:10.3390/toxins15090577_

Round 1
Reviewer 1 Report
The paper presents a review of the occurrence of selected microscopic fungi and their metabolites in maize. Conclusions from the conducted literature research are unambiguous and result from the available literature data. The discussion of the results based on the available literature is very detailed. The publications cited by the authors of the article are well selected. For the most part, the authors refer to the latest knowledge published in renowned scientific journals. I could not find any mistakes in the scientific aspect of the manuscript.
However, the authors did not avoid a few mistakes, which I will list below:
- A few punctuation problems are present in the manuscript. I suggest the Authors to double-check the text.
Author Response
The paper presents a review of the occurrence of selected microscopic fungi and their metabolites in maize. Conclusions from the conducted literature research are unambiguous and result from the available literature data. The discussion of the results based on the available literature is very detailed. The publications cited by the authors of the article are well selected. For the most part, the authors refer to the latest knowledge published in renowned scientific journals. I could not find any mistakes in the scientific aspect of the manuscript.
However, the authors did not avoid a few mistakes, which I will list below:
- A few punctuation problems are present in the manuscript. I suggest the Authors to double-check the text.
Response
We are very thankful for the reviewer’s comments and happy to hear such feedback. We have double-checked our manuscript and all the corrections are highlighted in yellow.

Reviewer 2 Report
Overall, this is a well written review publication and offers a good overview of the presence of some mycotoxins in maize. This manuscript shows rich content, providing a deep insight for some works: the study is within the journal’s scope, and I found it to be well-written, providing sufficient information. However, before publication some points need to be clarified.
My comments:
Line 52 – I think the authors should mention other fumonisins, such as FB2 and FB3, whose consumption leading to serious organism dysfunction is also widely reported.
Line 58 – both aflatoxins and fumonisns are very harmful both to human and animals. I think authors should briefly characterize the effects and symptoms of the mentioned mycotoxin intoxication (like disruption of endocrine and reproductive systems, disturbances in central and peripheral nervous system development and functioning behavioral changes and many more). It is especially important, in the light of recent finding showing that all mycotoxins are also a risk factor for organisms in prenatal life.
See:
1) Gao X, Xiao Z, Li C, Zhang J, Zhu L, Sun L, Zhang N, Khalil MM, Rajput SA, Qi D. Prenatal exposure to zearalenone disrupts reproductive potential and development via hormone-related genes in male rats. Food Chem Toxicol. 2018 Jun;116(Pt B):11-19.
2) Kras K, Rudyk H, Muszyński S, Tomaszewska E, Dobrowolski P, Kushnir V, Muzyka V, Brezvyn O, Arciszewski MB, Kotsyumbas I. Morphology and Chemical Coding of Rat Duodenal Enteric Neurons following Prenatal Exposure to Fumonisins. Animals (Basel). 2022 Apr 19;12(9):1055.
3) Supriya Ch, Reddy PS. Prenatal exposure to aflatoxin B1: developmental, behavioral, and reproductive alterations in male rats. Naturwissenschaften. 2015 Jun;102(5-6):26. doi: 10.1007/s00114-015-1274-7.
Table 1 – First replace “Turkey” with “Türkiye”. Second, because of its geographical location move this country to Asia. Also, Australia is both a continent and a country, so headline “other” is not needed.
Line 207, 225 – “in vivo” should be written in italics.
Author Response
Overall, this is a well written review publication and offers a good overview of the presence of some mycotoxins in maize. This manuscript shows rich content, providing a deep insight for some works: the study is within the journal’s scope, and I found it to be well-written, providing sufficient information. However, before publication some points need to be clarified.
Response
We are thankful for the reviewer’s feedback. We are happy to hear such critical evaluation and we agree with his opinion. We went again throughout the manuscript and all the corrections are highlighted in yellow.
My comments:
Line 52 – I think the authors should mention other fumonisins, such as FB2 and FB3, whose consumption leading to serious organism dysfunction is also widely reported.
Response
Thanks for the suggestion. We have mentioned them in the introduction.
Line 58 – both aflatoxins and fumonisns are very harmful both to human and animals. I think authors should briefly characterize the effects and symptoms of the mentioned mycotoxin intoxication (like disruption of endocrine and reproductive systems, disturbances in central and peripheral nervous system development and functioning behavioral changes and many more). It is especially important, in the light of recent finding showing that all mycotoxins are also a risk factor for organisms in prenatal life.
See:
1) Gao X, Xiao Z, Li C, Zhang J, Zhu L, Sun L, Zhang N, Khalil MM, Rajput SA, Qi D. Prenatal exposure to zearalenone disrupts reproductive potential and development via hormone-related genes in male rats. Food Chem Toxicol. 2018 Jun;116(Pt B):11-19.
2) Kras K, Rudyk H, Muszyński S, Tomaszewska E, Dobrowolski P, Kushnir V, Muzyka V, Brezvyn O, Arciszewski MB, Kotsyumbas I. Morphology and Chemical Coding of Rat Duodenal Enteric Neurons following Prenatal Exposure to Fumonisins. Animals (Basel). 2022 Apr 19;12(9):1055.
3) Supriya Ch, Reddy PS. Prenatal exposure to aflatoxin B1: developmental, behavioral, and reproductive alterations in male rats. Naturwissenschaften. 2015 Jun;102(5-6):26. doi: 10.1007/s00114-015-1274-7.
Response
Thanks so much for your comments. We have added the effects and symptoms of FB1 and AFB1 intoxication (please find it also in the manuscript >>page 2>>line 60 and line 69). Many thanks for suggesting those valuable references.
Table 1 – First replace “Turkey” with “Türkiye”. Second, because of its geographical location move this country to Asia. Also, Australia is both a continent and a country, so headline “other” is not needed.
Response
Thanks so much for your comments. We have corrected it (please find it also in the manuscript >>page 16>>Table 2).
Line 207, 225 – “in vivo” should be written in italics.
Response
Thanks so much for your comments. We have corrected it (please find it also in the manuscript >>page 18>>line 229 and line 248).

Reviewer 3 Report
This work provides an overview of the occurrence (and co-occurrences) of aflatoxins and fumonisins. A major suggestion taken from this work as a whole is that the co- occurrence of these toxins is increasing.
I think that the review should be published but there are a few key points that should be improved/added that I list below
Key Issues:
1) Table 1 showing studies that look for AF and FB toxins is useful. I would like the authors to make a similar Table to list all the of the studies they used for Figure 1 where the species of F. verticilloides and A. flavus were detected
2) Currently, figure 1 has limited value. Without a table that contains the studies in question its not possible to interpret. Its also not clear at all. These figure needs to be completely rethought and overhauled.
For example, it is not easy to determine the values of the various bar charts on the different regions. And, some of the positioning of the bar charts makes no sense. There is one chart in the arctic of Canada however I am sure no corn is growing there.
3) A major untapped resource for the co-occurence of mycotoxins in the DSM (formerly Biomin) mycotoxin surveys (https://www.biomin.net/mycotoxin-hub/)
This is extensive and multi year data that should be included in this work. Gathering the data from multiple years and sites will not be easy but again, it will greatly add value to this work.
Minor points:
On line 86, add reference: Battilani, Paola, et al. "Aflatoxin B1 contamination in maize in Europe increases due to climate change." Scientific reports 6.1 (2016): 24328. After this statement. (the reference is already in the review in other sections)
For figure 2, I’m having a difficult time figuring out what studies are used to derive this figure. I think as mentioned above, it would be useful to have a large table where the studies in question are listed with relevant information (year, location, fungal species searched for) There should be a table similar to Table 1.
L131. When the high concentrations are listed, please add the reference number.
L137, L140, L144W,145,hen the high concentrations are listed, please add the reference number.
Reference check
Can you verify through other sources the statement on L55 regarding deaths in Serbia. I cannot find anything in the references listed to support this.
Clarity issues
L29: It has been estimated that pre-harvest losses due to fungal plant diseases, nearly 10-20% of cultivated maize, can feed around 8.5% of the world population [2]. Rewrite this sentence for clarity.
L42: The International Agency for Research on Cancer (IARC) classified AFB1 as a carcinogenic agent (group 1 carcinogens) due to its potent hepatocellular carcinoma (HCC) in humans [12]. Rewrite this sentence for clarity.
L71. Figure 1 does not show the ‘incidence’ of the fungi, it shows the number of studies. The two are not necessarily linked.
L159: maize were selected as the threshold to observe and argue the collected data. Rewrite this sentence for clarity.
Author Response
This work provides an overview of the occurrence (and co-occurrences) of aflatoxins and fumonisins. A major suggestion taken from this work as a whole is that the co- occurrence of these toxins is increasing.
I think that the review should be published but there are a few key points that should be improved/added that I list below
Response
We are very thankful for the reviewer’s comments. We have considered all your comments and we changed the manuscript accordingly. Please, find below are our responses in details to all questions and comments that you suggested.
Key Issues:
1) Table 1 showing studies that look for AF and FB toxins is useful. I would like the authors to make a similar Table to list all the of the studies they used for Figure 1 where the species of F. verticilloides and A. flavus were detected
Response
Thanks so much for your comments. We have added a table (Table 1. Reported (co-)occurrences of A. flavus and F. verticillioides in maize worldwide.) to list all the of the studies they used for Figure 1 where the species of F. verticilloides and A. flavus were detected (please find it also in the manuscript >>page 4>>line 107).
2) Currently, figure 1 has limited value. Without a table that contains the studies in question its not possible to interpret. Its also not clear at all. These figure needs to be completely rethought and overhauled.
For example, it is not easy to determine the values of the various bar charts on the different regions. And, some of the positioning of the bar charts makes no sense. There is one chart in the arctic of Canada however I am sure no corn is growing there.
Response
Thanks so much for your comments. In order to make Figure 1 clear, we have added a table (Table 1. Reported the (co-)occurrence of A. flavus and F. verticillioides in maize and maize production worldwide.) to list all the of the studies they used for Figure 1 where the species of F. verticilloides and A. flavus were detected (please find it also in the manuscript >> page 4>>line 107).
3) A major untapped resource for the co-occurence of mycotoxins in the DSM (formerly Biomin) mycotoxin surveys (https://www.biomin.net/mycotoxin-hub/)
This is extensive and multi year data that should be included in this work. Gathering the data from multiple years and sites will not be easy but again, it will greatly add value to this work.
Response
Thanks so much for this great suggestion. We have went to the DSM link that you provided and we the link is not working. Perhaps they updated their website. Moreover, we would like that data collection was done from trusted and reliable sources such as PubMed and Web of Science, Scopus and google scholars .
Minor points:
On line 86, add reference: Battilani, Paola, et al. "Aflatoxin B1 contamination in maize in Europe increases due to climate change." Scientific reports 6.1 (2016): 24328. After this statement. (the reference is already in the review in other sections)
Response
Thanks so much for your comments. We have added it (please find it also in the manuscript >>page 2>>line 92).
For figure 2, I’m having a difficult time figuring out what studies are used to derive this figure. I think as mentioned above, it would be useful to have a large table where the studies in question are listed with relevant information (year, location, fungal species searched for) There should be a table similar to Table 1.
Response
Thanks so much for your comments. We have added a table (Table 1. Reported the (co-)occurrence of A. flavus and F. verticillioides in maize worldwide.) to list all the of the studies they used for Figure 2 where the species of F. verticilloides and A. flavus were detected (please find it also in the manuscript >> page 4>>line 107).
L131. When the high concentrations are listed, please add the reference number.
Response
Thanks so much for your comments. We have added it (please find it also in the manuscript >>page 14>>line 154 and line 160).
L137, L140, L144W,145,hen the high concentrations are listed, please add the reference number.
Response
Thanks so much for your comments. We have added it (please find it also in the manuscript >>page 14>>line 154 and line 160).
Reference check
Can you verify through other sources the statement on L55 regarding deaths in Serbia. I cannot find anything in the references listed to support this.
Response
Thanks so much for your comments. We have checked it and deleted it.
L29: It has been estimated that pre-harvest losses due to fungal plant diseases, nearly 10-20% of cultivated maize, can feed around 8.5% of the world population [2]. Rewrite this sentence for clarity.
Response
Thanks so much for your comments. We have rewritten this sentence (please find it also in the manuscript >>page 1>>line 44).
L42: The International Agency for Research on Cancer (IARC) classified AFB1 as a carcinogenic agent (group 1 carcinogens) due to its potent hepatocellular carcinoma (HCC) in humans [12]. Rewrite this sentence for clarity.
Response
Thanks so much for your comments. We have rewritten this sentence (please find it also in the manuscript >>page 2>>line 58).
L71. Figure 1 does not show the ‘incidence’ of the fungi, it shows the number of studies. The two are not necessarily linked.
Response
Thanks so much for your comments. We have changed it (please find it also in the manuscript >>page 2>>line 87).
L159: maize were selected as the threshold to observe and argue the collected data. Rewrite this sentence for clarity.
Response
Thanks so much for your comments. We have rewritten this sentence (please find it also in the manuscript >>page 17>>line 180).

Round 2
Reviewer 3 Report
I am satisfied with the authors response to my concerns and believe the article merits publication.